# Effect of Bio-Electro-Magnetic-Energy-Regulation (BEMER) Horse Therapy on Cardiopulmonary Function and Recovery Quality After Isoflurane Anesthesia in 100 Horses Subjected to Pars-Plana Vitrectomy: An Investigator-Blinded Clinical Study

**DOI:** 10.3390/ani14243654

**Published:** 2024-12-18

**Authors:** Olivier Brandenberger, Andrey Kalinovskiy, Jens Körner, Hermann Genn, Ralph Burger, Stephan Leser

**Affiliations:** 1Hanseklinik für Pferde, Karl-Benz-Straße 5-7, 27419 Sittensen, Germany; kalinovskiy333@gmail.com (A.K.); jens.koerner@hanseklinik.com (J.K.); stephan.leser@hanseklinik.com (S.L.); 2Pferdeklinik Mühlen, Münsterlandstraße 42, 49349 Steinfeld (Oldenburg), Germany; hj.genn@googlemail.com; 3Medical Expert Center, BEMER Int. AG, Austrasse 15, 9495 Triesen, Liechtenstein; kundenservice@bemer.service

**Keywords:** Bio-Electro-Magnetic-Energy-Regulation, equine, general anesthesia, pulsed electromagnetic fields, recovery, vasomotion

## Abstract

General anesthesia in horses is associated with high morbidity and mortality, and although several risk factors have been discovered, improvement, especially on hemodynamics and recovery, is subject to ongoing research. In order to investigate the effect of pulsed electromagnetic fields system BEMER (Bio-Electro-Magnetic-Energy-Regulation) on the hemodynamics and recovery quality of general anesthesia during ophthalmologic surgery, we divided 100 client-owned horses undergoing a routine procedure into two equal groups, and these were either treated with a BEMER therapy or placebo blanket. Hemodynamic parameters during surgery were recorded, recoveries were scored by a blinded observer, and the data were compared between the groups. The BEMER-therapy group demonstrated significantly higher recovery scores (indicating inferior recovery), while the hemodynamic data did not differ between the two groups. Therefore, an effect on recovery was detectable and may help in causing fewer complications surrounding necessary operative interventions in horses, although further trials are needed to optimize BEMER-therapy application during general anesthesia in the horse.

## 1. Introduction

Cardiovascular depression, hypoxemia, hypoventilation and eventful recoveries are common problems associated with general inhalation anesthesia in horses. Horses may injure themselves during the recovery phase, when they attempt to stand as they regain consciousness due to inappropriate muscle strength or poor coordination [1,2,3,4]. The mortality of healthy horses undergoing general anesthesia for elective cases ranges from 0.08% to 1.8%, depending upon study design [5,6,7,8,9]. Intraoperative cardiac arrest or postoperative cardiovascular collapse contribute to approximately one third of the deaths recorded. Another third is attributed to fractures and post-anesthesia myopathy. The latter is attributed to reduced intraoperative muscle perfusion and oxygen delivery [7,8,10]. While the duration of surgery was identified as a risk factor for prolonged recovery, several other factors like hypothermia or the selection and dosage of drugs also contribute to these risks [11,12,13,14,15].

Microcirculation in the blood vessels ensures transport in both directions between cells and vessels. Oxygen and nutrients are transported to the cell, and the proteins formed within the cell, together with metabolic end products, are transported out of the cells into the bloodstream [16,17]. One of the main criteria affecting blood perfusion is the diameter of the micro-vessels that are lined with smooth muscle cells. Changes in the diameter require wall movements which are achieved by the different contraction patterns of these smooth muscle cells. The smooth muscle cells in the large-caliber arterioles react to neural and humoral commands, whereas those in the small-caliber arterioles are under autorhythmic control [16,17]. In healthy humans, the repetition rate of these autorhythmically controlled vasomotions of the smallest precapillary and postcapillary vessels is approximately 3 times per minute. In seriously ill or older patients, as low as 1 vasomotion per 10 min has been described. Drugs (e.g., beta blockers) can only be used to influence vasomotion in the larger arterioles due to the lack of corresponding receptors in smaller vessels. As the smaller arterioles lack these receptors, vasomotion can only be stimulated by transferring physical energy [17,18].

Pulsed electromagnetic fields (PEMFs) are widely used in equine training and rehabilitation [19,20,21], despite the scarcity of literature data concerning their effects in horses. Bio-Electro-Magnetic-Energy-Regulation (BEMER) is a PEMF therapy with marked effects on microcirculation and autorhythmically controlled vasomotion [22,23,24], while the underlying mechanisms are not yet fully explored. The system creates a 10–100 μT pulsed electromagnetic field with half-wave-shaped sinusoidal intensity variations, which stimulates the body’s regulatory mechanisms for organ blood flow, especially in cases of limited regulatory ranges or dysfunctional perfusions such as those encountered during general anesthesia in the horse [3,4]. The defined physical energy transfer creates a physiological stimulus and influences the periodic contraction behavior of smooth muscle cells in the arteriolar vessel walls to produce a more efficient vasomotion pattern in regard to optimizing metabolic function [17]. We therefore applied BEMER-horse therapy in pilot trials on horses under general anesthesia to ascertain whether the effects of this therapy on microcirculation would increase perfusion and oxygenation of the muscles and thereby reduce post-anesthesia myopathy and influence recovery quality as other studies demonstrated similar effects or tendencies [25,26,27]. Creatine kinase as a marker of muscle damage as well as lactate, which elevates in case of hypoxemia, were identified as potential markers for the assessment of microcirculation issues [28].

Our goal was to quantify the influence of BEMER-horse therapy on cardiopulmonary function and recovery quality after general anesthesia. The authors perform several hundred routine pars-plana vitrectomies per year, using a standardized protocol for anesthesia, positioning, surgery procedure and surgical team. This protocol was therefore used to develop a study trial in which only one parameter would be changed to observe the effect on the quality of general anesthesia and its recovery. This parameter was either the use of the BEMER-horse therapy blanket (B+ group) or a placebo blanket (P− group). Due to its potential effects on vasomotion, we hypothesized that the B+ group would have lower blood lactate, higher perioperative blood pressure, better arterial oxygen tension and better recovery quality than the P− group while undergoing general anesthesia for pars-plana vitrectomy surgeries. 

## 2. Materials and Methods

This study was designed as a placebo-controlled, operator-blinded, randomized clinical study. According to German Landesamt für Verbraucherschutz und Lebensmittelsicherheit (LAVES) and the German Act of Prevention of Cruelty to Animals, the study was not classified as an animal experiment. The horses were treated in the same way as all the routine patients undergoing general anesthesia at the “Hanseklinik für Pferde” in Germany with the only difference being the application of BEMER-horse therapy blankets during the anesthesia process. Owners gave informed consent for their horse’s inclusion in the study. 

### 2.1. Animals and Study Design

The study was carried out on horses undergoing planned pars plana vitrectomy between August 2019 and January 2020. In all cases, the reason for vitrectomy was equine recurrent uveitis.

The horses were divided into an experimental group (BEMER-horse therapy, B+, *n* = 50) and a control group (Placebo, P−, *n* = 50). Only horses with no accompanying disorders and no additional surgical procedures were included in the study. The groups were operator blinded and a pre-filled list from 1 to 100 was randomly assigned to either group B+ or P− prior to the study using https://www.graphpad.com/quickcalcs/randomize2/ (accessed on 5 August 2019).

Two active BEMER-horse blankets (BEMER Horse-Set, BEMER Int. AG, Triesen, Liechtenstein) were used in group B+, whereas two inactivated BEMER-horse blankets were used in group P−. When the placebo blankets were switched on, all lights were activated but no energy flows were generated in the BEMER module. The operator would not be aware of any difference between the B+ and P− blankets. 

Two blankets were used for each horse: one underneath the horse and one on top, covering the shoulder, back and gluteus muscles (Figure 1). Both blankets were switched to level 3 immediately after placing the first portal for the pars-plana vitrectomy. The blankets were active for 15 min and switched off automatically. 

The basic module of the BEMER signal is a patent-protected BEMER pulse according to Prof. Dr. Wolf A. Kafka [29,30]. The pulse is described by following equation:y = (x^a^ × k × e^sin (x to the power of b)^) ÷ c + d

In this equation, y equals the value for amplitude of generated signal progression, x describes time, and a defines the parameter for timed adjustment of amplitude of each signal pulse with a value of 3. b is the number of superimposed pulses with a value of 3. Variable c characterizes the factor for amplitude adjustment with a value of 1 and d is the offset value with a value of 0, while k illustrates the factor for adjustment of an amplitude of the superimposed pulses with a value of 1.

The pulse is emitted by modules in the blanket, and Figure 2 shows the placement of the modules in the BEMER blanket for horses.

The horses were hospitalized and premedicated 3 days before the elective surgery. A preanesthetic examination was performed on the day of surgery in all cases. During the preanesthetic examination, venous blood was taken immediately before premedication. Complete blood count (CBC) and total protein (TP) were determined using an automated system (ProCyte Dx, IDEXX Laboratories Inc., Westbrook, ME USA). Surgery was performed under general anesthesia by the same surgeon (author SL) in all cases with standardized anesthetic procedures and drug dosage under monitoring of blood parameters as well as hemodynamic parameters.

### 2.2. Anesthesia and Instrumentation

The premedication consisted of the systemic administration of phenylbutazone 4.4 mg/kg bodyweight (bwt) (Phenylbutazon-Gel PH 100 mg/mL, CP-Pharma Handelsgesellschaft mbH, Burgdorf, Germany) and trimethoprim/sulfadiazine 5 mg/kg bwt/25 mg/kg bwt (Trimethosel-P 333.3 mg/mL Sulfadimethoxine 66.7 mg/mL, Trimethoprim, Selectavet Dr. Otto Fischer GmbH, Weyarn/Holzolling, Germany). Atropine eye drops (Atropin-POS 0.5%, Ursapharm GmbH, Saarbrücken, Germany) and an eye ointment that contained dexamethasone, neomycin sulfate and polymyxin B-sulfate (Isopto-Max, Novartis Pharma GmbH, Nürnberg, Germany) were administered topically three times a day. Twelve hours before surgery, the animals were deprived of food, but water was available ad libitum. 

One of three blinded anesthesiologists performed anesthesia following a standardized protocol. Immediately before surgery, a central venous catheter (Cavafix, B Braun Melsungen AG, Melsungen, Germany) was placed in the left or right jugular vein. The catheter placed on the side of the eye being operated was later used for premedication and infusion. Thirty minutes before surgery, acepromazine 0.03 mg/kg bwt (Tranquisol P 10 mg/mL, CP-Pharma Handelsgesellschaft mbH, Burgdorf, Germany) was administered intramuscularly. Xylazine 0.8 mg/kg bwt (Xylavet 20 mg/mL, CP-Pharma Handelsgesellschaft mbH, Burgdorf, Germany) and levomethadone 0.05 mg/kg bwt (L-Polamivet 2500 mg/7 mL, Intervet Deutschland GmbH, Unterschleißheim, Germany) were administered intravenously (IV) immediately before induction. Induction was performed with diazepam 0.05 mg/kg bwt (Ziapam, 5 mg/mL, Laboratoire TVM, Lempdes, France) and ketamine 2.5 mg/kg bwt (Ketamin 100 mg/mL, CP-Pharma Handelsgesellschaft mbH, Burgdorf, Germany).

The horses were supported during the induction phase with a head and tail rope. Horses were intubated, positioned identically in left or right lateral recumbency and connected to the anesthesia device for large animals (LAVC-2000D, Eickemeyer Medizintechnik für Tierärzte KG, Tuttlingen, Germany). Isoflurane (Isofluran CP 1 mg/mL, CP-Pharma Handelsgesellschaft mbH, Burgdorf, Germany) was administered in a mixture of oxygen and air. The inspiratory oxygen fraction (FIO_2_) was between 50% and 60%, and the end tidal isoflurane (FE’Iso) was initially 4%. Depending on the depth of anesthesia, the FE’Iso was reduced to 2–2.5% after 10 min. Mechanical ventilation was with a tidal volume (V_T_) between 8 and 15 mL/kg and a respiratory frequency (RF) between 4 and 6 breaths per minute. The V_T_ and RF were adjusted depending on the size of the horse and the end tidal carbo dioxide tension (P_E_’CO_2_). The aim was to obtain a P_E_’CO_2_ of 5.5–6.5 kPa (41–49 mmHg). Immediately after connecting the horses to the anesthesia system, a balanced crystalline solution infusion 10 mL/kg/h (lactated Ringer’s solution, B. Braun Vet Care, B Braun Melsungen AG, Melsungen, Germany) and 0.05% xylazine solution continuous rate infusion (CRI) at 2 mL/kg/h were administered for the duration of anesthesia, and a urinary catheter was also placed.

Anesthesia was monitored with an appropriate device (SurgiVet, Smiths Medical ASD, Inc., St. Paul, MN, USA) and included capnography, electrocardiography and invasive blood pressure measurement. For the invasive blood pressure measurement, an arterial cannula (VasoVet, B Braun Melsungen AG, Melsungen, Germany) was placed in the dorsal metatarsal artery (left or right, depending on the horse’s position) and connected to the anesthesia monitor by a pressure transducer (Smiths Medical, Smiths Medical ASD, Inc., St. Paul, MN, USA) positioned at the heart base and zeroed to atmospheric pressure. The anesthesia protocol recorded the heart rate (HF), systolic (SAP), diastolic (DAP) and mean (MAP) arterial blood pressure, BF, FE’Iso and V_T_ O_2_ every 5 min throughout anesthesia. The FE’Iso was adjusted depending on the MAP during surgery. 

Five to ten minutes before the end of surgery, the horses were weaned from mechanical ventilation by a gradual reduction in RF. Upon onset of the swallowing reflex or nystagmus, romifidine 0.08 mg/kg bwt (Sedivet 10 mg/mL, Boehringer Ingelheim Vetmedica GmbH, Ingelheim am Rhein, Germany) was administered intravenously by the anesthetist. At the onset of spontaneous breathing, the horses were moved to a darkened and padded recovery box. The horses were placed in the same recumbency as during surgery. A custom-made protective mask was placed over the head of the horse. Immediately after positioning the mask, the horses were extubated and a 12 mm ID nasotracheal tube was inserted. Recovery was unassisted for all horses and observed by the blinded operator.

### 2.3. Blood Samples and Blood Pressure Recording

Five venous blood samples were taken from the catheter placed for blood sampling. The CK value was determined twice and the lactate value was determined four times. Measurements were performed with a blood chemistry device (Catalyst One, IDEXX Laboratories Inc., Westbrook, ME, USA). 

The first CK sample was taken before premedication and approximately 15 min prior to induction. The first lactate sample was taken just after induction in the laying down box. The second lactate sample was taken just before the BEMER blankets were switched on 10 min after induction. The third lactate sample was collected after the horse was moved and lifted into the recovery box, approximately 10 min after surgery. The second CK sample and the fourth lactate sample were taken 15 min after recovery from anesthesia on the standing horse.

Two arterial blood gas samples were taken anaerobically during anesthesia from the arterial catheter and analyzed within 2 min. The first sample was collected before switching on the BEMER blanket. The second sample was collected immediately after the automatic switch-off of the BEMER blanket. An automatic, daily calibrated blood gas analyzing system (VetStat Electrolyte and Blood Gas Analyzer, IDEXX Laboratories Inc., Westbrook, ME, USA) was used. Blood gas analysis included an assessment of pH, partial pressures of arterial oxygen (PaO_2_) and carbon dioxide (PaCO_2_), arterial oxygen saturation (SaO_2_), base excess (BE) and plasma bicarbonate concentrations (HCO_3_)^−^.

MAP was measured continuously during general anesthesia and recorded for analyses just before the first blood gas collection and 10, 20 and 30 min after the first blood gas collection. 

### 2.4. Recovery

The 10-category scoring system first described by Donaldson et al. [31] was used to evaluate the recovery phase. Briefly, the categories include an evaluation of overall attitude, the activity in recumbency, movements to sternal recumbency, sternal phase, movements to stand up, strength, balance and coordination, knuckling, number of attempts to sternal recumbency and number of attempts to stand up. The total score ranged from 10 to 72 (best = 10; worst = 72) and scores > 72 were allowed according to the number of attempts to turn to sternal recumbency and number of attempts to stand up. Score was assessed by a blinded operator. 

### 2.5. Data Analysis

A power analysis was conducted to determine the required sample size. The power analysis was based on a 5% significance level and with 80% power. The recovery score, which was assumed to be normally distributed, was used as a primary outcome. In the pilot study (unpublished data), the recovery score mean difference was 6.1 with a standard deviation of 9.4. Sample size was therefore calculated to be a minimum of 78 animals. Concerning for possible lack of data or complications that may require deviation from the standardized protocol, a sample size of 100 animals was defined.

Data were analyzed in R, version 4.0 (R Core Team (2020). R: A language and environment for statistical computing. R Foundation for Statistical Computing, Vienna, Austria).

Mean and standard deviation for preoperative and postoperative CK values, blood gas values before and after the BEMER-horse blanket activation, lactate values and MAP measurements were collected. Each measurement was compared individually between the two groups by independent parametric examination using paired T-tests. The total recovery score was analyzed by applying an ordinary least squares regression analysis, controlling for the age and sex of the horse and for the anesthetist.

## 3. Results

All 100 horses completed the standardized study protocol, and full data sets were available.

### 3.1. Animals

The age (range 2–26 years), sex (mares *n* = 43, stallions *n* = 5, geldings *n* = 52) and breed (Warmblood *n* = 59, Islandic horse *n* = 11, Andalusian horse *n* = 6, Coldblood *n* = 6, Pony *n* = 6, Thoroughbred *n* = 4, Standardbred *n* = 3, Quarter horse *n* = 3, and Friesian horse *n* = 1) of the groups were comparable. All horses completed pars plana vitrectomy and anesthesia induction and maintenance was unremarkable in all horses. 

### 3.2. Blood Samples and Parameters of General Anesthesia

Pre- and postprocedural levels of CK, blood gas analyses, lactate and MAP were comparable at different timepoints and showed no statistical significance at any timepoint (Table 1).

### 3.3. Recovery

All horses recovered without major injury. Three horses showed hemorrhage in the operated eye after recovery. The total recovery score differed significantly between the groups (*p* = 0.00749). The mean score for the B+ group was 22.4 (standard deviation, SD 13.0), and the mean score for P− was 16.1 (SD 7.15) (Figure 3).

The regression analysis did not reveal any significant effect on the total recovery score due to the anesthetist or the age or sex of the horse (Table 2).

## 4. Discussion

Our study was focused on the effects of BEMER-horse therapy on general anesthesia and recovery quality. As BEMER-horse therapy influences microcirculation, we thought that this should increase perfusion and oxygenation of the muscles and thereby reduce post-anesthesia myopathy and improve recovery quality. We found a significant effect of the BEMER-horse therapy on the recovery quality; interestingly, the recovery was worse for the group receiving BEMER-horse therapy compared to that of the placebo group. 

Although to our knowledge, BEMER-horse therapy has not been investigated during general anesthesia of the horse, BEMER therapy effects have been investigated in several human studies. Piatkowski et al. [32] were able to demonstrate a beneficial effect of BEMER therapy on fatigue in human patients suffering from multiple sclerosis. After 12 weeks of daily treatment, the treatment group showed significantly lower fatigue levels than the placebo group. They suggested that the weak, low frequency pulsed electromagnetic fields influenced energy metabolism and oxygen supply on a cellular basis and stimulated the microcirculation, which led to fewer signs of neuronal fatigue. In contrast to our study, the human participants did not receive medical treatment or even general anesthesia. The way in which BEMER-horse therapy affects horses that are under the influence of anesthetic drugs has remained unknown until now. 

One possible explanation for this finding is that in the B+ group, the anesthetic drugs were delivered and absorbed more effectively, and a deeper anesthesia plane was reached with the same anesthetic protocol compared to the P− group. A deeper anesthesia in the B+ group was also reflected with regard to blood pressure in our study. Blood pressure in the B+ group tended to be lower over time than in the P− group. It is generally recognized by the veterinary anesthetic community that the depth of anesthesia influences recovery quality: the deeper the anesthesia, the longer the duration of the excitation phase during recovery, thus promoting incoordination during this phase. We believe that the improved microvascularization obtained after BEMER-horse therapy may have led to better absorption and distribution of the anesthetic drugs, a deeper plane of anesthesia in the B+ horses and therefore a higher recovery score. However, depth of anesthesia was not measured in this study, and prospective studies are warranted to evaluate the immediate effect of BEMER-horse therapy on the drug absorption and depth of anesthesia in horses.

Measurements were made of CK values, blood gas, lactate and blood pressure to evaluate the cardiovascular state of the horse during general anesthesia. No significant differences could be found between the B+ and P− group for any of these factors. It is possible that due to the study design, the duration of general anesthesia was too short to influence these parameters. Furthermore, BEMER-therapy was applied for only 15 min to each horse. In the human study by Benedetti et al. [22], the treatment was applied for 20 min per day for 10 consecutive days. Although the authors were able to document an effect on the complex regional pain syndrome, they also recommended that the treatment should be performed over a longer period in order to enhance the effects [22]. A treatment of longer duration or even prior to induction and after recovery might influence the measured parameters, and further studies are therefore warranted.

The present study has several limitations. The general anesthesia for vitrectomy cases is deep in order to permit turning and manipulation of the horse’s eye. General anesthesias, where the horse is kept in a lighter plane, used for other routine surgeries (e.g., arthroscopy or orthopedic surgeries), might react differently to intra-operative BEMER therapy. This should be studied in future trials. The recoveries were classified by just one veterinary surgeon. Although it was always the same person and an objective protocol [31] was used, a second or more observers would have been beneficial to reduce possible bias [33,34,35] to evaluate inter-observational agreement. Furthermore objective, non-observer-dependent scoring systems may be included in future trials as well [9].

## 5. Conclusions

We examined the effect of BEMER-horse therapy on 100 horses undergoing general anesthesia for pars plana vitrectomy and were able to demonstrate that BEMER-horse therapy did significantly influence recovery, which is the most threatening periprocedural period regarding complications. These findings might be due to enhanced vasomotion and increased blood distribution. More studies are warranted to examine the effects of BEMER-horse therapy, especially on the depth of anesthesia in relation to adjuvant stimulation of microcirculation to support anesthesia management.

## Figures and Tables

**Figure 1 animals-14-03654-f001:**
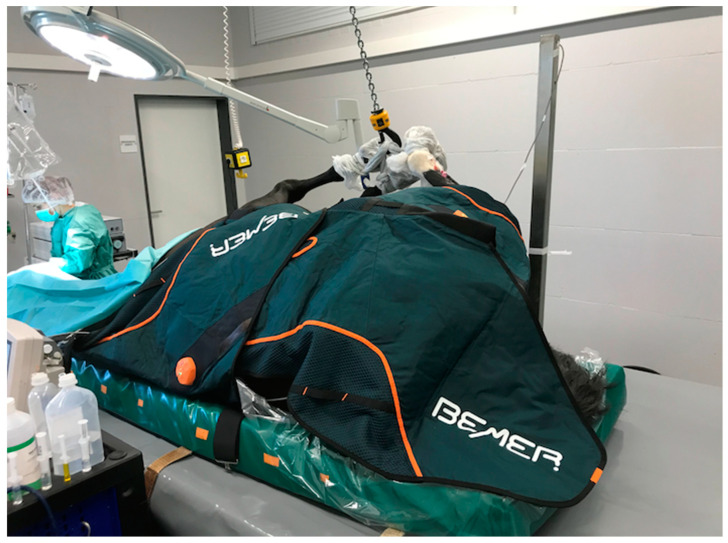
Application of BEMER-horse therapy blankets during general anesthesia.

**Figure 2 animals-14-03654-f002:**
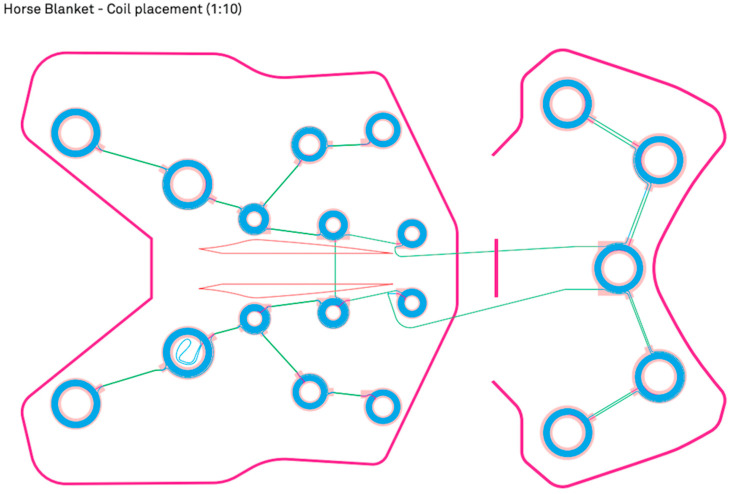
Diagram of module placement in the BEMER-horse therapy blanket. The pink line represents the outline. The blue circles represent the modules.

**Figure 3 animals-14-03654-f003:**
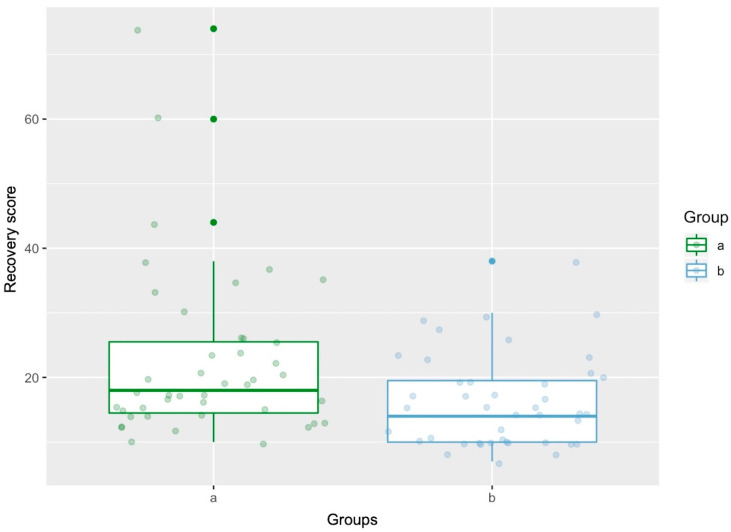
Boxplot demonstrating recovery scores of BEMER-horse therapy group (a) and placebo-treatment group (b) with 95% confidence intervals.

**Table 1 animals-14-03654-t001:** Mean values (with standard deviation) during general anesthesia for the BEMER-horse-therapy group (B+) and placebo group (P−) with results of significance tests.

Measured Value	Timepoint	BEMER-Therapy Group (B+)	Placebo Group (P−)	*p*-Value
CK (µkat/L)	Before induction	164.4 (39.8)	177.0 (87.3)	0.24
	After recovery	217.9 (130.1)	231.3 (154.8)	0.58
PaO_2_ (mmHg)	Before activation	137.8 (62.6)	160.3 (87.0)	0.17
	After activation	153.3 (95.6)	177.8 (93.3)	0.26
PaCO_2_ (mmHg)	Before activation	56.6 (7.3)	57.4 (8.1)	0.62
	After activation	62.2 (10.5)	62.7 (11.5)	0.83
MAP (mmHg)	Before activation	86.3 (23.1)	93.2 (23.3)	0.20
	+10 min	75.9 (19.9)	79.8 (20.2)	0.45
	+20 min	66.5 (15.1)	70.6 (19.5)	0.37
	+30 min	63.3 (13.1)	68.4 (18.2)	0.23
Lactate (mmol/L)	After induction	1.5 (0.7)	1.4 (0.5)	0.28
	Before activation	1.6 (0.4)	1.5 (0.4)	0.68
	10 min after surgery	1.5 (0.5)	1.7 (0.7)	0.10
	15 min after recovery	1.8 (1.6)	2.2 (2.1)	0.37

CK = creatine kinase, PaO_2_ = partial pressure of arterial oxygen, PaCO_2_ = partial pressures of arterial oxygen, MAP = mean arterial pressure.

**Table 2 animals-14-03654-t002:** Regression analysis using an ordinary least squares regression analysis controlling for horse age and sex, and for the anesthetist.

Value	Estimate (Standard Deviation)	Significance
Intercept B+	21.97 (4.95)	
Group P−	−5.36 (2.05)	*p* < 0.05
Anesthetist	0.76 (2.07)	*p* > 0.05
Age	−0.1 (0.19)	*p* > 0.05
Sex (female)	−1.68 (4.58)	*p* > 0.05
Sex (gelding)	−0.59 (4.63)	*p* > 0.05
R^2^	0.11	
Adj. R^2^	0.05	
Number of observations	94	
Root mean squared Error (RMSE)	8.63	

## Data Availability

The raw data supporting the conclusions of this article will be made available by the authors on request.

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
