# Peer review of "Effect of Bio-Electro-Magnetic-Energy-Regulation (BEMER) Horse Therapy on Cardiopulmonary Function and Recovery Quality After Isoflurane Anesthesia in 100 Horses Subjected to Pars-Plana Vitrectomy: An Investigator-Blinded Clinical Study"

_animals, 2024, doi:10.3390/ani14243654_

Round 1

Reviewer 1 Report

Comments and Suggestions for Authors

Simple Summary and Abstract

You state significantly different recovery scores – please clarify, that the recovery of the BEAMER group was worse. The Summary and abstract are otherwise misleading. 

4. Discussion:

Line 305 – 310

Please change to less certain phrasing: may have led or similar

Line 324

Change wording; as it is the horse or patient and not the surgeries that might react differently 

Line 326/327

You state classified by one person but previously you say three anaesthetists. Please clarify. 

Author Response

Thank you very much for your critical comments. We adapted the document according to your comments and we believe that it has improved a lot. Thank you!

Please find the comments for every changing attached. 

Reviewer 2 Report

Comments and Suggestions for Authors

This manuscript presents a prospective clinical study of the effect of BEMER therapy during general anesthesia. According to my knowledge, this is the first study to examine the effect of BEMER-horse therapy. The finding that BEMER-horse therapy did significantly infuence recovery is intringuing given that the exact effects of BEMER-horse therapy are neither known nor proven. The fact that recovery was indeed worse does not lessen the fact that a statistically sound relationship has been proven. Given the recent hype around some veterinarians, technicians and horse owners to use Bio-Electro-Magnetic-Energy-Regulation and the frequent use of pulsed electromagnetic fields in equine training and rehabilitation, this is important work and should be published.

Specific comments:

Line 17: divided instead of diveded

LIine 106: at the "Hanseklinik" instead of at th "Hanseklinik"

Line 116: were included in the study instead of included the study

Author Response

Thank you very much for your comments. We changed everything according to your comments, please see the document attached. 
